# Exosomal miRNAs: Key Regulators of the Tumor Microenvironment and Cancer Stem Cells

**DOI:** 10.3390/ijms26199323

**Published:** 2025-09-24

**Authors:** Shuangmin Wang, Sikan Jin, Jidong Zhang, Xianyao Wang

**Affiliations:** 1Department of Immunology, Zunyi Medical University, Zunyi 563000, China; wangshuangmin220@foxmail.com (S.W.); jkk24sa@163.com (S.J.); jdzhang0851@163.com (J.Z.); 2Key Laboratory of Cancer Prevention and Treatment of Guizhou Province, Zunyi Medical University, Zunyi 563000, China

**Keywords:** exosomes, miRNA, cancer, CSCs

## Abstract

Exosomes are lipid bilayer vesicles approximately 30–150 nm in diameter that serve as key mediators of intercellular communication. By transporting diverse bioactive molecules, including proteins and nucleic acids, they play a crucial role in tumor initiation and progression. Among their functional cargo, exosomal microRNAs (miRNAs) are central to epigenetic regulation and intercellular signaling, significantly influencing tumor biology. This review provides a comprehensive overview of the multifaceted roles of exosomal miRNAs in remodeling the tumor microenvironment (TME) and regulating cancer stem cells (CSCs). Specifically, exosomal miRNAs modulate various immune cells (such as macrophages, T cells, and NK cells) as well as cancer-associated fibroblasts (CAFs), thereby promoting immune evasion, angiogenesis, epithelial–mesenchymal transition (EMT), and metastatic progression. At the same time, they enhance CSC stemness, self-renewal, and therapeutic resistance, ultimately driving tumor recurrence and dissemination. Furthermore, exosome-mediated miRNA signaling acts as a critical force in malignant progression. Finally, we discuss the clinical potential of exosomal miRNAs as diagnostic and prognostic biomarkers, therapeutic targets, and vehicles for targeted drug delivery, highlighting their translational value and future directions in cancer research.

## 1. Introduction

Cancer remains a major global health threat with a highly complex pathogenesis, in which therapeutic resistance and recurrent metastasis pose persistent clinical challenges. According to the 2022 Global Cancer Statistics, approximately 20 million new cancer cases and 9.7 million related deaths were reported worldwide [1]. Although significant progress has been made in diagnostic and therapeutic approaches, treatment efficacy continues to be limited by TME-mediated immune evasion, drug resistance, and metastasis [2]. In recent years, the CSC hypothesis has provided a critical theoretical framework for understanding tumor heterogeneity, treatment failure, and disease recurrence. CSCs, characterized by self-renewal, multipotency, and strong resistance to therapies, are considered key drivers of tumor initiation, progression, and relapse [3]. Intercellular communication plays a central role in the dynamic regulation of the TME and the maintenance of CSC functions. Exosomes, a major subclass of extracellular vesicles, act as vital carriers of intercellular communication by transporting bioactive molecules such as proteins and nucleic acids, and are deeply involved in cancer development and progression [4]. Among their diverse cargo, exosomal miRNAs represent a crucial class of epigenetic regulators. By transferring across cells, they induce gene silencing and modulate signaling pathways, thereby influencing tumor proliferation, invasion, immune microenvironment remodeling, and CSC regulation [5]. Recent studies have shown that exosomal miRNAs not only regulate various TME components—including cancer-associated fibroblasts, tumor-associated macrophages, and other immune cells—to promote angiogenesis, matrix remodeling, and immune suppression [6,7,8], but also play critical roles in sustaining CSC stemness, self-renewal, and resistance mechanisms [9,10,11]. These findings highlight the multifaceted regulatory functions of exosomal miRNAs in malignant tumor progression. Therefore, systematically elucidating the roles and mechanisms of exosomal miRNAs in the TME and CSCs is essential for advancing our understanding of tumor recurrence and therapeutic resistance. Moreover, it provides a theoretical foundation and translational perspective for developing novel strategies such as exosomal miRNA-based liquid biopsy tools and combinational therapies. This review aims to summarize the diverse biological functions of exosomal miRNAs in the TME and CSCs, with a focus on their critical roles in cancer progression and treatment resistance, and to explore their clinical potential and future research directions.

## 2. Biogenesis and miRNA Sorting of Exosomes

Exosomes are extracellular vesicles containing various components such as proteins, nucleic acids, and lipids. They are widely present in biological fluids and participate in functions including intercellular information transmission, immune responses, and tissue repair. Additionally, they can serve as carriers for targeted tumor delivery, enhancing therapeutic efficacy. Notably, due to their natural biocompatibility and targeting ability, exosomes have emerged as ideal carriers for novel drug delivery systems, demonstrating significant advantages in targeted tumor therapy [12]. This process begins with the internalization of the plasma membrane: under clathrin-mediated endocytosis, the plasma membrane invaginates to form early endosomes. Subsequently, under the precise regulation of the endosomal sorting complex required for transport (ESCRT) system, the early endosomal membrane buds inward to form multiple intraluminal vesicles (ILVs), gradually developing into mature multivesicular bodies (MVBs) [13] The ESCRT system, through a cascade reaction of its core components (ESCRT-0 to ESCRT-III) in collaboration with ATPases such as VPS4, accomplishes the selective sorting of cargo molecules (e.g., transmembrane proteins, nucleic acids) and vesicle formation [14,15]. The fate determination of MVBs is a critical regulatory node in exosome generation: on one hand, MVBs can fuse with lysosomes and be degraded by hydrolytic enzymes such as cathepsins, a process regulated by GTPases like Rab7 [16]; on the other hand, mediated by GTPases such as Rab27a/b and Rab35, as well as SNARE proteins (e.g., VAMP7, Syntaxin-4), MVBs migrate directionally to the plasma membrane and fuse with it, releasing ILVs into the extracellular space via exocytosis, thereby forming exosomes [17]. The release of exosomes is a tightly regulated process involving the coordinated action of multiple Rab GTPases, SNARE proteins, and cytoskeletal components. This ensures their spatiotemporally precise secretion, enabling exosomes to mediate critical biological functions, including intercellular communication, material transport, and regulatory signaling [17].

The sorting of miRNAs into exosomes is an exceptionally complex and finely regulated process, with current research highlighting several key mechanisms: lipid microdomain-dependent sorting, in which exosomal membranes enriched in cholesterol, sphingolipids, and tetraspanins provide a structural basis for miRNA incorporation via lipid rafts, and where neutral sphingomyelinase 2 (nSMase2) was the first protein shown to regulate this process, as Kosaka et al. [18] demonstrated that inhibiting nSMase2 significantly reduces miRNA secretion while its overexpression enhances release, a finding later confirmed in mouse breast cancer models where nSMase2 overexpression increased exosome production and the levels of miR-16 and miR-210, with opposite effects observed upon nSMase2 suppression [19]; RNA-binding protein-mediated sorting, in which RNA-binding protein selectively recognize specific miRNA sequences or structural motifs, including hnRNPA2B1, which binds 3′-end GGAG and GGCU motifs and regulates sorting via ubiquitination [20], Y-box binding protein 1 (YBX1), which promotes miR-133 incorporation through its cold shock domain [21,22], and Argonaute 2 (Ago2), a functional miRNA effector whose knockout markedly decreases exosomal levels of miR-142-3p, miR-150, and miR-451 [23]; and 3′-end modification-dependent sorting, evidenced by widespread 3′ uridylation of urinary and B cell-derived exosomal miRNAs, suggesting that hydrophobic 3′ modifications and motifs like GGAG play critical roles in directing miRNA incorporation into exosomes [24]. These sorting mechanisms are finely tuned by cell type and dynamic microenvironmental conditions; however, due to technical limitations such as suboptimal exosome isolation purity and low sensitivity in detecting low-abundance miRNAs, a comprehensive understanding of exosomal miRNA sorting remains incomplete, and future studies integrating multi-omics approaches with functional validation are needed to fully elucidate the molecular mechanisms and biological significance of this process.

## 3. Cancer and Cancer Stem Cells

Cancer stem cells (CSCs) represent a distinct subpopulation within tumor tissues that possess stem cell-like properties including self-renewal capacity, multi-lineage differentiation potential, and tumorigenic ability, thereby being regarded as the “seed cells” responsible for tumor initiation and progression [25]. The seminal discovery in CSC research occurred in 1997 when Bonnet and Dick’s research team first successfully identified and isolated leukemia stem cells (LSCs) from acute myeloid leukemia using CD34^+^/CD38^−^ surface markers, marking a groundbreaking advancement in the field [26]. Subsequent progress in tumor biology has enabled researchers to identify CSC populations in various solid tumors (including breast cancer, glioma, hepatocellular carcinoma, gastric cancer, colorectal cancer, and ovarian cancer) through specific biomarkers such as CD44, CD133, and ALDH1 [27]. Notably, the expression profiles of CSC surface markers exhibit remarkable tumor-type specificity: ALDH^+^, CD44^+^, and CD133^+^ are predominantly expressed in head and neck cancer CSCs [28]; CD200^+^ and CD166^+^ are highly expressed in colorectal CSCs [29,30]; while pancreatic CSCs specifically overexpress markers including CD44/CD24 and epithelial-specific antigens [31]. Although CSCs typically constitute only 0.01–2% of the total tumor cell population, they play a pivotal role in tumor initiation, progression, metastasis, and recurrence [32]. Furthermore, CSCs frequently reside in a quiescent state within the cell cycle while overexpressing various drug resistance-associated molecules (such as ABC transporters and DNA damage repair enzymes), thereby serving as crucial mediators of therapeutic resistance. Mare et al. [33] demonstrated that paclitaxel-pretreated MCF7 breast cancer cells exhibited significantly enhanced stem cell properties, as evidenced by increased tumor sphere formation capacity, providing direct evidence for the central role of CSCs in chemoresistance. Recent studies have further revealed that beyond their remarkable self-renewal capacity, CSCs exhibit unique “phenotypic plasticity”—the ability to undergo epithelial–mesenchymal transition (EMT) or dedifferentiation in response to microenvironmental cues such as hypoxia and inflammatory cytokines. This adaptability leads to the generation of diverse tumor cell subpopulations. Such plasticity not only substantially contributes to tumor heterogeneity but also plays a critical role in metastasis and treatment resistance [34]. At the molecular level, CSCs maintain their stemness and regulate vital biological processes (including survival, proliferation, and differentiation) through aberrant activation of key signaling pathways (such as Wnt/β-catenin, Notch, and PI3K/AKT/mTOR) accompanied by specific epigenetic modifications (including DNA methylation and histone modifications) [35]. It is important to emphasize that these processes are closely linked to the dynamic changes within the TME. Exosomal miRNAs act on multiple components of the TME or directly regulate CSC stemness, phenotypic plasticity, and therapy resistance, thereby promoting sustained tumor progression and treatment evasion [36]. Thus, exosomal miRNAs represent a critical regulatory hub between tumor cells and CSCs, driving malignant evolution and providing a molecular basis for tumor recurrence and metastasis.

## 4. Exosomal miRNA-Mediated Tumor Microenvironment Remodeling and Malignant Progression

Exosomal miRNAs, as key mediators of intercellular communication, can transmit signaling molecules over both local and long distances through a stable circulatory transport system, playing a critical role in tumor initiation and progression. Within the TME, exosomal miRNAs promote malignancy through multiple mechanisms: first, by remodeling the cellular composition and signaling networks of the TME, inducing functional changes in CAFs and immune cells, thereby enhancing angiogenesis and immune suppression to facilitate tumor immune evasion; second, by mediating EMT, which increases tumor cell invasiveness and metastatic potential; and third, by transferring drug resistance-related molecules and regulating metabolic reprogramming to support tumor cell survival and proliferation. Collectively, exosomal miRNAs establish a multilayered regulatory network within the TME, exacerbating tumor progression and therapeutic resistance (see Figure 1 and Table 1).

The list of exosomal miRNAs presented in this table was compiled through a comprehensive search of PubMed and Web of Science databases up to September 2025. The search strategy employed a combination of the following keywords: (exosome OR extracellular vesicle) AND (microRNA OR miRNA OR miR) AND (cancer OR tumor OR carcinoma OR neoplasm) AND (immune OR cancer stem cell). Studies were included if they: (1) identified miRNAs specifically enriched in or secreted by cancer cell-derived exosomes and investigated their functions in immune or cancer stem cell regulation; (2) provided experimental validation of exosomal miRNA expression (e.g., via qRT-PCR, sequencing); and (3) were published in English. Relevant miRNAs from the selected studies were then extracted and summarized.

### 4.1. Exosomal miRNA-Mediated Immune Regulation and Immune Evasion

The TME comprises cancer cells, CAFs, immune cells, vascular endothelial cells, extracellular matrix, and various signaling molecules, forming a highly complex and dynamically evolving ecosystem [104]. Exosomal miRNAs act as central mediators of intercellular communication, reshaping the TME through diverse molecular mechanisms to enhance immunosuppressive characteristics and promote immune evasion [105], as summarized in Figure 2. Specifically, exosomal miRNAs can activate CAFs, drive matrix remodeling and pro-tumor factor secretion, and regulate the polarization of tumor-associated macrophages (TAMs), thereby coordinating the functions of multiple immune and stromal cell types. Functioning as essential messengers and molecular hubs linking tumor cells with their microenvironment, exosomal miRNAs play a pivotal role in driving immune escape and tumor progression.

#### 4.1.1. Exosomal miRNA Regulation of CAFs

CAFs are major stromal components of the TME in solid tumors and play a pivotal role in tumor initiation and progression. Emerging evidence indicates that exosomal miRNAs regulate CAF activation and function through multiple molecular mechanisms, thereby promoting the formation of an immunosuppressive TME. For example, Wang et al. reported that colorectal cancer-derived exosomal miR-146a-5p and miR-155-5p activate CAFs via the JAK2-STAT3/NF-κB pathway, enhancing the secretion of inflammatory cytokines such as IL-6, TGF-β, and CXCL12 [37]. In gastric cancer, exosomal miR-27a induces fibroblast reprogramming toward a CAF phenotype [38], while miR-496 upregulates IL-33 to enhance the pro-tumorigenic properties of CAFs, promoting gastric cancer cell proliferation, EMT, migration, and invasion [39]. In breast cancer, exosomal miR-185-5p, miR-652-5p, and miR-1246 induce normal fibroblasts to acquire a CAF-like phenotype [40], and miR-370-3p activates fibroblasts via the CYLD/NF-κB signaling pathway, facilitating tumor progression [41]. Additionally, lung cancer-derived exosomal miR-142-3p promotes CAF formation through non-canonical TGF-β signaling [42], and melanoma-derived exosomal miR-155 induces angiogenic CAFs by suppressing SOCS1 and activating the JAK2/STAT3 pathway [43]. Notably, exosomal miR-375 from Merkel cell carcinoma (MCC) can induce fibroblast polarization by inhibiting RBPJ and p53, further highlighting the broad and diverse roles of exosomal miRNAs in modulating CAF function across different tumor types [44]. Exosomal miRNAs also stimulate CAFs to secrete a variety of inflammatory cytokines and growth factors, reinforcing their pro-tumorigenic effects. For instance, in hepatocellular carcinoma, exosomal miR-1247-3p targets B4GALT3 to induce inflammatory gene expression in CAFs and activate the β1-integrin/NF-κB pathway, enhancing tumor-promoting activity [45]. Taken together, exosomal miRNAs mediate CAF activation and functional remodeling through multiple signaling pathways across diverse cancers, profoundly influencing the tumor immune microenvironment and malignant progression.

#### 4.1.2. Exosomal miRNA Regulation of Macrophages

TAMs are a major component of tumor-infiltrating immune cells and play a critical role in shaping the TME. In response to different cytokine stimuli, TAMs can polarize into two functionally distinct subtypes: M1 macrophages, which exhibit anti-tumor activity, and M2 macrophages, which promote tumor growth and metastasis by remodeling an immunosuppressive TME [106,107]. Extensive studies have shown that exosomal miRNAs can be internalized by macrophages and, by targeting specific signaling pathways or gene expression, suppress M1 functions while driving M2 polarization, thereby creating an immunosuppressive microenvironment and enhancing pro-angiogenic and tissue-repair functions. For example, epithelial ovarian cancer-derived exosomal miR-222-3p induces M2 polarization by suppressing SOCS3 and activating the STAT3 pathway, promoting EOC proliferation and metastasis [46]; under hypoxic conditions, pancreatic cancer-derived exosomal miR-301a-3p drives M2 polarization via PTEN inhibition and PI3Kγ pathway activation [47]; hepatocellular carcinoma exosomal miR-21-5p induces M2 polarization through activation of the RhoB/MAPK pathway, enhancing TGF-β and IL-10 secretion [48]; and melanoma-derived exosomal miR-125b-5p promotes a pro-tumor TAM phenotype while enhancing macrophage survival [49]. Collectively, tumor-derived exosomal miRNAs regulate macrophage phenotype and function, shaping an immunosuppressive TME, facilitating tumor immune evasion and progression, and providing potential targets for therapeutic intervention.

#### 4.1.3. Exosomal miRNA Regulation of DC and NK Cells

Dendritic cells (DCs) are the most potent antigen-presenting cells in the body, capable of activating naïve T cells by processing and presenting antigens, and thus play a central role in initiating and sustaining immune responses [108]. Exosomal miRNAs can be delivered to DCs, interfering with their maturation and antigen-presenting functions, leading to DC dysfunction or a shift toward a tolerogenic phenotype [109]. Mechanistically, this can involve suppression of costimulatory molecule expression, downregulation of MHC class II molecules, or induction of immunosuppressive cytokines, thereby weakening T cell activation and promoting immune evasion. For example, in pancreatic cancer, exosomal miR-212-3p targets RFXAP to reduce HLA class II expression, impairing DC function [50,51]; miR-let-7i modulates multiple key molecules, including IL-6, IL-17, TGF-β, SOCS1, and TLR4, suppressing DC-mediated immune responses [52]; and miR-203 inhibits DC maturation by downregulating TLR4 expression [53]. Other studies have shown that various miRNAs regulate DC survival and lifespan via YWHAZ and Bcl2 signaling pathways [110]. Exosomal miRNAs also modulate DC immunoregulatory functions—for instance, miR-17-5p suppresses TNF-α and IL-12 secretion while promoting IL-10 production, inducing a tolerogenic DC phenotype that inhibits T cell activation and promotes Treg expansion [54].

Natural killer (NK) cells, as key effectors of innate immunity, play a critical role in tumor surveillance and clearance [111]. Exosomal miRNAs can inhibit NK cell function through multiple mechanisms. In breast cancer, high levels of miR-20a downregulate the NKG2D ligands MICA/MICB, impairing NK cell recognition and cytotoxicity and facilitating immune evasion and metastasis [55]. In hepatocellular carcinoma, miR-17-5p suppresses RUNX1, leading to reduced NKG2D expression and compromised NK cell cytotoxicity [56], while liver cancer-derived exosomal miR-92b inhibits the expression of the NK activation marker CD69, diminishing NK cell killing activity [57].

#### 4.1.4. Exosomal miRNA Regulation of Myeloid-Derived Suppressor Cells

Myeloid-derived suppressor cells (MDSCs) are a heterogeneous population of immature myeloid cells with potent immunosuppressive activity that protect tumor cells by dampening immune responses, thereby playing a critical role in the TME [112]. Exosomal miRNAs can drive the differentiation of immature myeloid cells (IMCs) into MDSCs while inhibiting their maturation into antitumor immune cells, and can also enhance the immunosuppressive capacity of MDSCs, ultimately promoting tumor progression [113]. For instance, Jiang et al. demonstrated that miR-9 and miR-181a target SOCS3 and PIAS3, respectively, activating the JAK/STAT pathway to promote early MDSC generation and the establishment of an immunosuppressive microenvironment [114]. In melanoma, exosomal miR-125a-5p modulates myeloid cell phenotypes through the NF-κB pathway, leading to MDSC accumulation and suppression of T cell proliferation [115]. In hypoxic oral squamous cell carcinoma, exosomal miR-21 targets PTEN to upregulate PD-L1, activating MDSCs and impairing γδ T cell function [116]. Similarly, in hypoxic glioma, exosomal miR-10a regulates MDSC differentiation by targeting RORA and the NF-κB pathway, while miR-21 enhances their activation through the PTEN/STAT3/Akt pathway [117]. In GC, exosomal miR-107 promotes MDSC expansion and activation by targeting DICER1 and PTEN, highlighting its potential as a therapeutic target [118]. Notably, in lung cancer, exosomal miR-21a suppresses PDCD4, which not only accelerates tumor growth but also boosts MDSC expansion, further underscoring the broad role of exosomal miRNAs in shaping an immunosuppressive TME [119]. Collectively, these findings demonstrate that exosomal miRNAs serve as key mediators of tumor–MDSC interactions and immune evasion, offering promising avenues for tumor immunotherapy.

#### 4.1.5. Exosomal miRNA Regulation of T Cells

CD8^+^ T cells, also known as cytotoxic T lymphocytes (CTLs), are the central effector cells of adaptive immunity that recognize tumor-associated antigens and mediate specific cytotoxicity, playing a pivotal role in antitumor immune defense [120]. Tumor-derived exosomal miRNAs can be internalized by T cells, where they regulate key signaling pathways and transcription factors to modulate T-cell activation, proliferation, differentiation, and effector functions. For example, colorectal cancer exosomal miR-424 suppresses the costimulatory signal by downregulating CD28, thereby impairing full T-cell activation and facilitating immune evasion [121]; leukemia exosomal miR-19a-3p targets SLC6A8 to inhibit creatine uptake, weakening CD8^+^ T-cell function [122]. In addition, exosomal miRNAs promote immunosuppression through regulation of the PD-1/PD-L1 axis: gastric cancer exosomal miR-1246 inhibits GSK3β to stabilize PD-L1 and induce CD8^+^ T-cell apoptosis [123]; in triple-negative breast cancer, miR-20a-5p suppresses CD8^+^ T-cell function and mediates resistance to PD-1 blockade [124]; and cancer stem cell-derived exosomal miR-17-5p upregulates PD-L1 by targeting SPOP, thereby suppressing antitumor immunity [125]. CD4^+^ T cells (Th cells) orchestrate immune responses through cytokine secretion, with Th1-derived IFN-γ serving as a key mediator of antitumor immunity. Exosomal miR-let-7d inhibits Th1 cell proliferation and IFN-γ production, promoting immune tolerance [126]. Moreover, studies in miR-155-deficient mice have shown accelerated tumor growth in syngeneic lymphoma models, accompanied by reduced proportions of IFN-γ^+^ CD4^+^ T cells and impaired IFN-γ mRNA expression in CD8^+^ T cells, indicating that miR-155 plays a crucial role in antitumor immunity by enhancing IFN-γ responses [127].

### 4.2. Promotion of Tumor Angiogenesis

Angiogenesis supplies tumor cells with essential nutrients and oxygen while also providing routes for metastatic dissemination, making it a key event in cancer progression [128]. Exosomal miRNAs contribute to this process by transmitting pro-angiogenic signals, regulating gene expression, and activating critical signaling pathways. For instance, hepatocellular carcinoma-derived exosomal miR-103 targets endothelial junction proteins such as VE-cadherin, p120-catenin, and zonula occludens-1, thereby increasing vascular permeability and promoting metastasis [58], while nasopharyngeal carcinoma exosomal miR-23a enhances endothelial proliferation, migration, and tube formation by suppressing TSGA10 [59]. Hypoxia, a major inducer of tumor angiogenesis, upregulates leukemia-derived exosomal miR-210, which promotes endothelial angiogenesis by targeting Ephrin-A3 and activating the VEGF/VEGFR2 pathway [60], and also increases tissue inhibitor of metalloproteinases-1 (TIMP-1) expression, thereby activating the PI3K/AKT/HIF-1 pathway to drive miR-210 transcription and amplify its pro-angiogenic effects [61]. Exosomal miR-21 induces STAT3 activation and VEGF upregulation, promoting angiogenesis and malignant transformation of human bronchial epithelial cells [62], while hypoxic lung cancer-derived exosomal miR-23a suppresses prolyl hydroxylase (PHD) and tight junction protein ZO-1, enhancing vascular permeability and facilitating tumor transendothelial migration [63]. Similarly, ovarian cancer SKOV-3 cell-derived small extracellular vesicles (sEVs) enriched in miR-141-3p activate the JAK-STAT3 pathway to promote angiogenesis [64], mesenchymal stem cell (MSC)-derived exosomal miR-100 regulates the mTOR/HIF-1α/VEGF axis to facilitate breast cancer angiogenesis [65], and exosomal miR-9 in nasopharyngeal carcinoma inhibits migration and angiogenesis by modulating the PDK/Akt pathway [66].

### 4.3. Promotion of EMT and Metastasis

Exosomal miRNAs promote tumor growth and dissemination by modulating the expression of oncogenes and tumor suppressor genes in recipient cells [129]. For example, colorectal cancer-derived exosomal miR-200b directly binds the 3′ untranslated region of p27 to suppress its expression, thereby accelerating tumor growth [67], while in hepatocellular carcinoma, exosomal miR-584 targets TAK1 and inhibits the MAPK-JNK tumor-suppressive pathway to drive progression and metastasis [68]. Similarly, glioblastoma exosomal miR-148a suppresses CADM1 and activates STAT3 signaling, markedly enhancing proliferation, invasion, and metastatic potential [69]. Exosomal miRNAs also interfere with cell cycle and apoptosis regulation; for instance, miR-1246 downregulates CCNG2 to disrupt cell cycle control and promote breast cancer proliferation and chemoresistance [70], whereas breast cancer-derived exosomal miR-128 modulates Bax and other Bcl-2 family proteins to inhibit apoptosis and augment metastatic capacity [71]. Collectively, these mechanisms highlight the multifaceted role of exosomal miRNAs in promoting malignant progression. A central step in this process is EMT, which endows tumor cells with migratory, invasive, and metastatic abilities, characterized by the downregulation of epithelial markers such as E-cadherin, the upregulation of mesenchymal markers such as vimentin and N-cadherin, and enhanced motility [130]. Exosomal miRNAs are pivotal drivers of EMT: tumor-derived exosomal miR-21 upregulates Snail and alters the vimentin/E-cadherin ratio to markedly enhance migration and invasion [72]; renal cancer stem cell-derived exosomal miR-19b-3p suppresses PTEN to induce EMT and facilitate metastasis [73]; hypoxia-induced bone marrow mesenchymal stem cell exosomal miR-193a-3p, miR-210-3p, and miR-5100 activate STAT3 signaling to trigger EMT and increase invasiveness in lung cancer [74]; and colorectal cancer exosomal miR-335-5p targets RASA1 to accelerate EMT, invasion, and metastatic spread [75]. Together, these findings underscore exosomal miRNAs as crucial mediators of EMT and potent drivers of tumor migration, invasion, and distant metastasis.

### 4.4. Roles of Exosomal miRNAs in Drug Resistance and Metabolic Reprogramming

Exosomal miRNAs play pivotal roles in both chemotherapy resistance and metabolic reprogramming, two interlinked processes that collectively drive malignant progression and therapeutic resistance. In mediating chemoresistance, exosomal miRNAs can endow tumor cells with drug-resistant phenotypes through diverse mechanisms. For instance, Liang et al. reported that colorectal cancer-derived exosomal miR-21 suppresses PTEN and hMSH2, thereby enhancing resistance to 5-fluorouracil [76], while in breast cancer, exosomal miR-221/222 downregulate p27 and ERα to promote tamoxifen resistance [77]. Additional mechanisms include facilitating drug efflux, blocking apoptosis, and inducing EMT: exosomal miR-155 inhibits PTEN and Fas ligand to disrupt apoptotic signaling [78,79], and both miR-21 and miR-27a downregulate E-cadherin while upregulating mesenchymal markers, thereby inducing EMT and strengthening chemoresistance in pancreatic and breast cancers [80,81,82]. In hepatocellular carcinoma, exosomal miR-92a-3p derived from highly metastatic cells activates the PTEN/Akt pathway to induce EMT and metastatic capacity in low-metastatic counterparts [83].

In terms of metabolic reprogramming, exosomal miRNAs profoundly influence tumor cell energy metabolism by modulating key metabolic enzymes and signaling pathways, thereby promoting aberrant processes such as glycolysis and lipid synthesis. For example, miR-155 indirectly upregulates hexokinase 2 (HK2) by repressing miR-143, enhancing glycolytic flux [84]. Exosomal miRNAs also regulate lipid metabolism: miR-21 elevates lipogenic enzymes such as FASN and ACC1, thereby promoting proliferation and migration in non-small cell lung cancer [85]. In breast cancer, exosomal miR-1304-3p activates cancer-associated adipocytes (CAAs), contributing to disease progression in African American patients [86], whereas tumor-derived miR-204-5p reprograms white adipose tissue through leptin signaling, enhancing lipolysis and fueling tumor progression [87]. Furthermore, matrix stiffness-tuned exosomal miRNAs can establish a glucose-enriched pre-metastatic niche in the lungs, providing a favorable metabolic microenvironment for distant metastasis [131]. At the signaling level, exosomal miRNAs orchestrate metabolic reprogramming via PI3K/AKT/mTOR and AMPK pathways: miR-21 activates PI3K/AKT to promote glycolysis [88,89], while miR-451 has been implicated in AMPK-mediated energy regulation that supports tumor growth [90]. Collectively, these findings demonstrate that exosomal miRNAs not only enhance chemoresistance by regulating drug-resistance-associated genes but also remodel tumor metabolism through glycolysis, lipid metabolism, and microenvironmental metabolic reprogramming. Their dual function in drug resistance and metabolic adaptation confers a significant survival advantage to tumor cells and underpins therapeutic failure.

## 5. Roles of Exosomal miRNAs in Cancer Stem Cells

Exosomal miRNAs can precisely regulate the activity of oncogene promoters and the expression of tumor suppressor genes, thereby significantly influencing multiple functional mechanisms, biological behaviors, and phenotypic characteristics of CSCs, as shown in Figure 3. Their core regulatory roles are mainly manifested in modulating the self-renewal capacity, differentiation direction, invasive and metastatic potential of CSCs, as well as the development of therapeutic resistance.

### 5.1. Maintenance of CSC Stemness Characteristics and Self-Renewal

The Wnt, Notch, and Hedgehog signaling pathways are key regulators of tumor stemness [132]. Exosomal miRNAs modulate these pathways to regulate the sphere-forming ability and tumorigenicity of CSCs, thereby maintaining and promoting their stem-like phenotype. Wu et al. found that miR-483-5p promotes proliferation, invasion, and self-renewal of gastric cancer stem cells by activating the Wnt/β-catenin signaling pathway [91]. Under the regulation of the transcription factor HIF-1α, miR-1275 activates both the Wnt/β-catenin and Notch pathways, enhancing stemness in lung adenocarcinoma cells and promoting tumorigenicity, recurrence, and metastasis [92]. Yang et al. [93] reported that chemotherapy activates the EZH2/STAT3 axis in breast cancer cells, leading to secretion of exosomes containing miR-378a-3p, which upon uptake by surviving cells activate Wnt/Notch pathways to enhance stemness and confer drug resistance. Exosomal miR-454 from breast cancer cells maintains CSC stemness in ovarian cancer by activating the PRRT2/Wnt axis [94]. M2 macrophage-derived exosomal miR-27a-3p promotes liver cancer CSC characteristics, proliferation, and tumorigenicity in vivo via activation of the TXNIP pathway [95]. Additionally, exosomal miRNAs target stemness-related gene expression to promote CSC traits. Chemotherapy induces breast cancer cells to secrete exosomes carrying multiple miRNAs such as miR-9-5p, miR-195-5p, and miR-203a-3p, which target ONECUT2, induce CSC phenotypes, and upregulate stemness-associated gene expression [96]. miR-328-3p is significantly upregulated in ovarian CSCs, maintaining stemness by targeting and suppressing DNA damage-binding protein 2. Concurrently, low reactive oxygen species levels reduce ERK signaling activity, favoring enhanced miR-328 expression and CSC maintenance [97].

### 5.2. Mediation of CSC Therapy Resistance

CSCs are considered key contributors to tumor drug resistance. Increasing evidence shows that CSC-derived exosomes deliver miRNAs to non-CSCs, promoting chemoresistance through multiple mechanisms including apoptosis regulation, enhanced drug efflux, and induction of drug resistance in chemo-sensitive cells [133,134,135]. CSC-derived exosomes reprogram recipient cells’ cell cycle and apoptosis-related genes to facilitate resistance. Yin et al. [98] demonstrated that under hypoxia, miR-30b-3p in glioma CSC-derived exosomes targets RHOB, reducing apoptosis and promoting resistance to temozolomide. Zhuang et al. [99] reported that exosomal miR-146a-5p from cancer-associated fibroblasts regulates cell cycle and apoptosis pathways, maintaining bladder CSC stemness and enhancing chemoresistance, thus establishing a pro-resistant tumor microenvironment niche. Yang et al. [100] showed that exosomes from gemcitabine-resistant pancreatic CSCs transfer miR-210 to drug-sensitive cells, conferring resistance accompanied by upregulation of MDR1, YB-1, BCRP, and activation of the mTOR pathway, suggesting miR-210 mediates resistance by promoting drug efflux in non-CSC subpopulations. miR-221 suppresses the tumor suppressor QKI by targeting its mRNA 3′-UTR, leading to aberrant activation of downstream stemness pathways, which promotes tumorigenicity and chemoresistance of colorectal CSCs [101]. CSC-derived exosomes transmit miRNAs and other genetic materials that enable horizontal transfer of resistance to sensitive cells. Santos et al. experimentally confirmed that miR-155 secreted by breast CSC exosomes downregulates C/EBP-β and inhibits TGF-β, C/EBP-β, and FOXO3a expression, inducing EMT and chemoresistance in sensitive cells [102]. miR-485-5p specifically suppresses keratin 17 (KRT17) expression, thereby regulating integrin-mediated FAK/Src/ERK signaling and β-catenin nuclear translocation, ultimately affecting oral CSC stemness maintenance and chemoresistance phenotypes [103].

## 6. Translational Applications of Exosomal miRNAs in Cancer Diagnosis and Therapy

### 6.1. Potential Diagnostic and Prognostic Biomarkers

Exosomal miRNAs, owing to their remarkable stability, high tissue and disease specificity, and widespread presence in various body fluids, are regarded as promising diagnostic and prognostic biomarkers with significant clinical potential [136]. They have demonstrated important value in the detection and evaluation of malignancies, neurological disorders, and cardiovascular diseases. Wu et al. [137] examined the expression profiles of eight serum and serum-derived exosomal miRNAs (including miR-21-5p and miR-141-3p) in healthy individuals, patients with benign lung lesions, and patients with early-stage (NSCLC, stage I/II). Their findings revealed that serum exosomal miRNAs outperformed serum miRNAs as biomarkers for early NSCLC diagnosis, and the combination of both further enhanced diagnostic accuracy. Wang et al. [138] reported that the combination of CA15-3 in exosomes and serum miR-1910-3p served as an effective biomarker, improving the reliability of breast cancer diagnosis. In prostate cancer patients, plasma exosomal miR-141-5p was upregulated, while miR-125a-5p was downregulated, suggesting their potential as diagnostic indicators [139]. In oral squamous cell carcinoma (OSCC), elevated plasma exosomal miR-130a levels were identified as independent predictors of overall survival and recurrence-free survival [140]. Liu et al. [141] demonstrated that serum exosomal miR-106b-3p was significantly increased in metastatic CRC compared to non-metastatic cases, indicating its potential as a prognostic biomarker and therapeutic target. Additionally, urinary exosomal miR-155-5p, miR-15a-5p, and miR-21-5p were markedly elevated in bladder cancer patients compared to healthy controls, supporting their utility as non-invasive diagnostic indicators [142].

### 6.2. Potential Therapeutic Targets

#### 6.2.1. Targeted Delivery of Exosomal miRNAs to Tumor Cells

Exosomes can be engineered to encapsulate specific tumor-suppressive miRNAs for precise delivery to target cells, offering novel therapeutic strategies. For instance, miR-122 loaded into MSC-derived exosomes effectively targeted hepatocellular carcinoma cells, exerting anti-tumor effects [143]. miR-375, a tumor suppressor inversely correlated with EMT, was delivered via tumor-derived exosomes in the form of mimics by Rezaei et al. [144], thereby suppressing migration and invasion in colon cancer cells. In gastric cancer, CAF-derived exosomes loaded with miR-139 mimics significantly inhibited tumor cell proliferation and metastasis [145],. Similarly, miR-205 mimics transfected into bone marrow mesenchymal stem cell-derived exosomes suppressed RHPN2 expression, impeding prostate cancer progression [146]. In a recent study, Cui et al. [147] employed electroporation to load miR-486-5p into bone marrow mesenchymal stem cell-derived exosomes and deliver them specifically to tumor sites. The results demonstrated that this strategy effectively inhibited glycolysis and stemness in colorectal cancer cells by targeting NEK2, thereby significantly suppressing tumor growth and progression. In addition to delivering tumor-suppressive miRNAs, exosomes can serve as carriers for miRNA inhibitors or antagonists to block oncogenic miRNA activity. This can be achieved using anti-miRNA oligonucleotides or inhibitors that interfere with transcription, processing, or stability. For example, Wang et al. [148] encapsulated a miR-21 inhibitor into engineered exosomes and delivered it to gastric cancer cells, achieving greater inhibitory effects and lower cytotoxicity compared to conventional transfection methods. Liang et al. [76] co-loaded the chemotherapeutic drug 5-FU and a miR-21 inhibitor (miR-21i) into exosomes for targeted delivery to Her2-positive colorectal cancer and 5-FU-resistant cell lines. This approach restored the function of tumor suppressor PTEN and DNA repair protein hMSH2, inducing cell cycle arrest and apoptosis. Furthermore, transfection of an miRNA-BART1-5p antagonist into exosomes significantly inhibited angiogenesis in tumor tissues and induced apoptosis [149]. Collectively, these studies highlight the substantial potential of exosomes in miRNA-based targeted cancer therapy, offering novel strategies and insights for precision oncology.

#### 6.2.2. Targeted Delivery of Exosomal miRNAs to Cancer Stem Cells

The maintenance of CSC stemness is highly dependent on the regulation of multiple signaling pathways within the tumor microenvironment. As essential intercellular communication vehicles, exosomes can deliver specific miRNAs to CSCs, modulating various stemness-related signaling pathways and thereby altering their biological properties to suppress tumor growth and progression. In recent years, the advent of engineered exosomes has provided a feasible and efficient strategy for targeted miRNA delivery to CSCs. Epithelial cell adhesion molecules are one of the classic CSC markers. Studies have shown that engineering exosomes to specifically recognize and target Epithelial cell adhesion molecule -expressing liver CSCs, while loading them with β-catenin-specific siRNA, can effectively block activation of the Wnt/β-catenin signaling pathway and significantly inhibit CSC proliferation [150]. Similarly, tumor-suppressive miRNAs can be loaded into exosomes for CSC targeting. For example, Alessia et al. [151] demonstrated that exosomes derived from human liver stem cells (HLSCs) can deliver miR-145 and miR-200 to renal CSCs, inducing apoptosis and markedly suppressing proliferation, sphere-forming capacity, and invasiveness. SOX9, a critical regulator of CSC stemness maintenance, self-renewal, and tumor progression, has emerged as an important therapeutic target for CSC-directed interventions. Wu et al. [152] reported that bone marrow mesenchymal stem cell-derived exosomes carrying miR-145-5p specifically target SOX9 in non-small cell lung cancer, thereby significantly impairing CSC stemness maintenance and inhibiting tumor cell proliferation, migration, and invasion. In addition, Lang et al. [153] demonstrated that transducing miR-124a into MSCs enables the generation of miR-124a-enriched exosomes (Exo-miR-124a), which can be effectively delivered to glioblastoma stem cells (GSCs). In vitro, Exo-miR-124a markedly reduced GSC viability and clonogenic potential, while in an orthotopic GSC xenograft mouse model, systemic administration of Exo-miR-124a resulted in 50% of tumor-bearing mice achieving long-term tumor-free survival, indicating complete tumor eradication. Beyond tumor-suppressive miRNAs, engineered exosomes can also be utilized to deliver inhibitors of pro-stemness miRNAs (anti-miRs) to attenuate CSC tumorigenicity. For instance, Naseri et al. [154] employed electroporation to load anti-miR-142-3p into exosomes for targeted delivery to tumor sites, effectively suppressing endogenous miR-142-3p expression and function. This intervention disrupted signaling pathways associated with breast CSC tumorigenicity, significantly reducing CSC tumorigenic potential and impeding malignant progression of breast cancer.

## 7. Conclusions and Future Perspectives

Exosome-derived miRNAs, as key mediators of intercellular communication, play crucial roles in cancer and CSC biology by reshaping the tumor microenvironment, regulating CSC stemness maintenance, and modulating therapeutic resistance. They are deeply involved in tumor initiation and progression, metastasis, and treatment resistance. Moreover, the detectability and specificity of exosomal miRNAs in liquid biopsies provide a solid foundation for their use as biomarkers for early cancer diagnosis, therapeutic response assessment, and prognostic prediction. In recent years, strategies employing engineered exosomes as miRNA delivery vehicles have achieved promising results in preclinical studies, offering new therapeutic avenues for targeting CSCs and overcoming drug resistance. However, several challenges remain in this field. First, the mechanisms underlying exosomal miRNA biogenesis, selective packaging, and secretion are not yet fully elucidated, limiting their precise application in early disease diagnosis. Second, the techniques for exosome isolation and purification, miRNA loading efficiency, and targeted in vivo delivery still require optimization to ensure safety and controllability. Furthermore, the functions of exosomal miRNAs within CSCs and the tumor microenvironment are highly context-dependent and exhibit spatiotemporal specificity, adding complexity to clinical translation. Future research should focus on the following directions: Elucidating the dynamic regulatory networks of exosomal miRNAs in CSC stemness maintenance and tumor progression; Developing efficient, standardized techniques for exosome isolation and quantitative miRNA detection to improve reproducibility and accuracy in clinical testing; Advancing the application of engineered exosomes in miRNA-targeted delivery by optimizing loading efficiency and targeting specificity to achieve CSC-specific eradication; Integrating single-cell omics, multi-omics, and spatial transcriptomics to uncover the key mechanisms by which exosomal miRNAs regulate tumor heterogeneity and resistance evolution; Accelerating the clinical translation of exosomal miRNA-based diagnostic and therapeutic strategies, grounded in rigorous safety evaluations. In summary, research on exosome-derived miRNAs in cancer and CSCs is evolving from fundamental mechanistic studies toward clinical application. With deeper insights into their molecular mechanisms and the refinement of engineering-based delivery technologies, exosomal miRNAs are poised to become integral components of future precision oncology, offering innovative solutions to the challenges of tumor recurrence, metastasis, and therapeutic resistance.

## Figures and Tables

**Figure 1 ijms-26-09323-f001:**
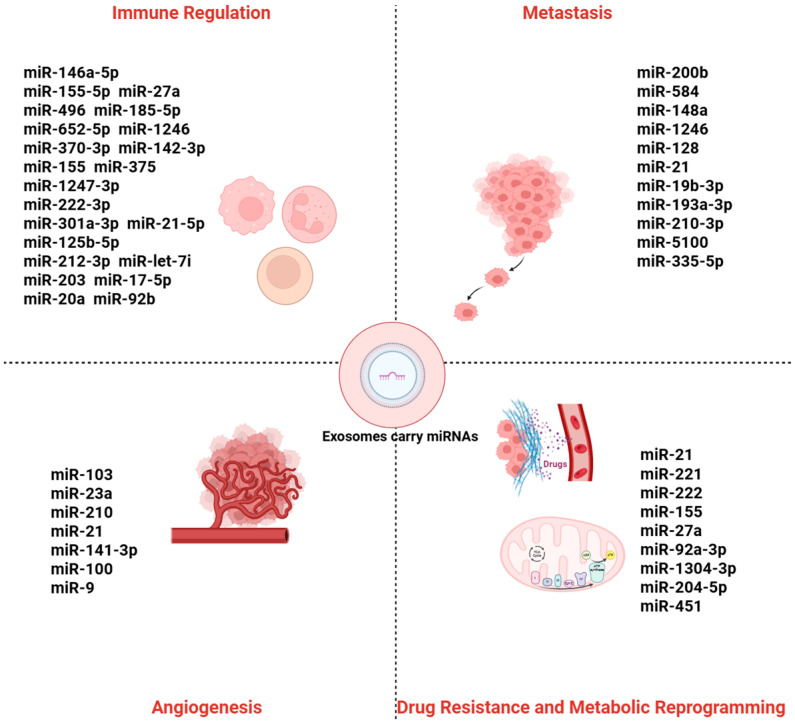
Roles of exosomal miRNAs in tumor progression. Exosomal miRNAs can accelerate tumor metastasis, induce angiogenesis, enhance drug resistance and tumor metabolic reprogramming, and contribute to the formation of a tumor-promoting and immunosuppressive microenvironment.

**Figure 2 ijms-26-09323-f002:**
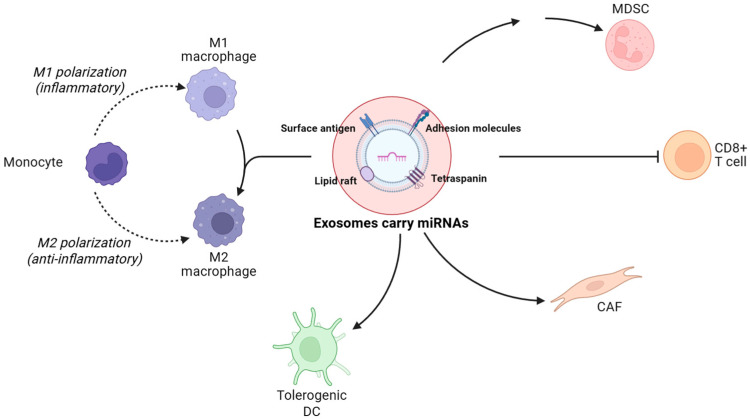
TME consists of cancer cells, CAFs, immune cells, vascular endothelial cells, extracellular matrix, and signaling molecules. Exosomal miRNAs act as key mediators by activating CAFs, driving matrix remodeling, and regulating TAM polarization, thereby enhancing immunosuppression, promoting immune evasion, and facilitating tumor progression.

**Figure 3 ijms-26-09323-f003:**
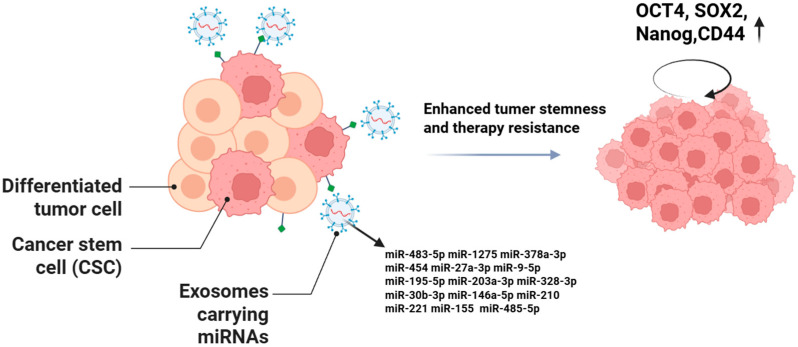
Exosomal miRNAs can enhance the stemness of CSCs, which is primarily manifested by increased expression of stemness markers, enhanced tumorsphere-forming ability, and strengthened self-renewal capacity.

**Table 1 ijms-26-09323-t001:** Tumor-Derived Exosomal miRNAs and Their Roles in the Tumor Microenvironment and Cancer Progression.

Functional Module	Exosomal miRNA	Source	Function	Mechanism	Reference
CAF Regulation	miR-146a-5p, miR-155-5p	Colorectal cancer	CAF activation	JAK2-STAT3/NF-κB pathway; increased IL-6, TGF-β, CXCL12 secretion	[37]
	miR-27a	Gastric cancer	Induces CAF phenotype	Fibroblast reprogramming	[38]
	miR-496	Gastric cancer	Enhances pro-tumorigenic CAF traits	Upregulates IL-33, promotes proliferation, EMT, migration, invasion	[39]
	miR-185-5p, miR-652-5p, miR-1246	Breast cancer	Induces CAF-like phenotype	—	[40]
	miR-370-3p	Breast cancer	CAF activation	CYLD/NF-κB signaling pathway	[41]
	miR-142-3p	Lung cancer	Promotes CAF formation	Non-canonical TGF-β signaling	[42]
	miR-155	Melanoma	Induces angiogenic CAFs	Suppresses SOCS1, activates JAK2/STAT3	[43]
	miR-375	Merkel cell carcinoma	Fibroblast polarization	Inhibits RBPJ and p53	[44]
	miR-1247-3p	Hepatocellular carcinoma	CAF activation	Targets B4GALT3; β1-integrin/NF-κB pathway	[45]
TAM Regulation	miR-222-3p	Epithelial ovarian cancer	M2 polarization	Suppresses SOCS3; activates STAT3	[46]
	miR-301a-3p	Pancreatic cancer (hypoxia)	M2 polarization	PTEN inhibition; PI3Kγ activation	[47]
	miR-21-5p	Hepatocellular carcinoma	M2 polarization	RhoB/MAPK pathway; enhances TGF-β and IL-10 secretion	[48]
	miR-125b-5p	Melanoma	Promotes pro-tumor TAM phenotype	Enhances macrophage survival	[49]
DC & NK Cell Regulation	miR-212-3p	Pancreatic cancer	Impairs DC function	Targets RFXAP; reduces HLA class II expression	[50,51]
	miR-let-7i	Multiple cancers	Suppresses DC-mediated immune response	Regulates IL-6, IL-17, TGF-β, SOCS1, TLR4	[52]
	miR-203	Multiple cancers	Inhibits DC maturation	Downregulates TLR4	[53]
	miR-17-5p	Multiple cancers	DC immunoregulation	Suppresses TNF-α and IL-12; promotes IL-10 secretion	[54]
	miR-20a	Breast cancer	Inhibits NK cells	Downregulates NKG2D ligands MICA/MICB	[55]
	miR-17-5p	Hepatocellular carcinoma	Impairs NK cell cytotoxicity	Suppresses RUNX1; reduces NKG2D expression	[56]
	miR-92b	Liver cancer	Diminishes NK cell activity	Inhibits CD69 expression	[57]
Angiogenesis	miR-103	Hepatocellular carcinoma	Promotes angiogenesis	Targets VE-cadherin, p120-catenin, ZO-1	[58]
	miR-23a	Nasopharyngeal carcinoma	Enhances endothelial proliferation, migration, tube formation	Suppresses TSGA10	[59]
	miR-210	Leukemia (hypoxia)	Stimulates angiogenesis	Targets Ephrin-A3; activates VEGF/VEGFR2	[60,61]
	miR-21	Multiple cancers	Promotes angiogenesis	Activates STAT3; upregulates VEGF	[62]
	miR-23a	Hypoxic lung cancer	Increases vascular permeability	Suppresses PHD and ZO-1	[63]
	miR-141-3p	Ovarian cancer (SKOV-3)	Promotes angiogenesis	Activates JAK-STAT3 pathway	[64]
	miR-100	Mesenchymal stem cells	Facilitates angiogenesis	Regulates mTOR/HIF-1α/VEGF axis	[65]
	miR-9	Nasopharyngeal carcinoma	Modulates angiogenesis	PDK/Akt pathway	[66]
EMT & Metastasis	miR-200b	Colorectal cancer	Promotes tumor growth	Binds p27 3′UTR; suppresses expression	[67]
	miR-584	Hepatocellular carcinoma	Promotes progression and metastasis	Targets TAK1; inhibits MAPK-JNK tumor-suppressive pathway	[68]
	miR-148a	Glioblastoma	Enhances proliferation, invasion, metastasis	Suppresses CADM1; activates STAT3	[69]
	miR-1246	Breast cancer	Promotes proliferation and chemoresistance	Downregulates CCNG2; disrupts cell cycle	[70]
	miR-128	Breast cancer	Inhibits apoptosis; enhances metastasis	Modulates Bax and Bcl-2 family proteins	[71]
	miR-21	Multiple cancers	Induces EMT; increases migration and invasion	Upregulates Snail; alters vimentin/E-cadherin ratio	[72]
	miR-19b-3p	Renal CSC	EMT induction and metastasis	Suppresses PTEN	[73]
	miR-193a-3p, miR-210-3p, miR-5100	MSC (hypoxia)	Triggers EMT; increases invasiveness	Activates STAT3 signaling	[74]
	miR-335-5p	Colorectal cancer	Accelerates EMT, invasion, metastasis	Targets RASA1	[75]
Drug Resistance & Metabolic Reprogramming	miR-21	Colorectal cancer	5-FU resistance	Suppresses PTEN, hMSH2	[76]
	miR-221/222	Breast cancer	Tamoxifen resistance	Downregulates p27 and ERα	[77]
	miR-155	Multiple cancers	Anti-apoptosis; promotes chemoresistance	Inhibits PTEN and Fas ligand	[78,79]
	miR-21, miR-27a	Pancreatic, breast cancer	EMT induction; enhances chemoresistance	Downregulates E-cadherin; upregulates mesenchymal markers	[80,81,82]
	miR-92a-3p	Hepatocellular carcinoma	EMT and metastasis	Activates PTEN/Akt pathway	[83]
	miR-155	Multiple cancers	Enhances glycolysis	Suppresses miR-143; upregulates HK2	[84]
	miR-21	NSCLC	Promotes lipogenesis	Upregulates FASN, ACC1	[85]
	miR-1304-3p	Breast cancer	Activates cancer-associated adipocytes	Reprograms adipose tissue; promotes tumor progression	[86]
	miR-204-5p	Tumor	White adipose tissue metabolic reprogramming	Modulates leptin signaling; enhances lipolysis	[87]
	miR-21	Multiple cancers	Glycolysis promotion	Activates PI3K/AKT pathway	[88,89]
	miR-451	Multiple cancers	Energy metabolism regulation	AMPK signaling	[90]
CSC Stemness & Therapy Resistance	miR-483-5p	Gastric CSC	Maintains stemness; self-renewal	Activates Wnt/β-catenin pathway	[91]
	miR-1275	Lung adenocarcinoma	Enhances stemness; promotes metastasis	Activates Wnt/β-catenin and Notch pathways	[92]
	miR-378a-3p	Breast cancer (post-chemotherapy)	Enhances stemness; chemoresistance	Activates Wnt/Notch pathways	[93]
	miR-454	Breast cancer	Maintains CSC stemness	Activates PRRT2/Wnt axis	[94]
	miR-27a-3p	M2 macrophage	Promotes CSC proliferation and tumorigenicity	Activates TXNIP signaling	[95]
	miR-9-5p, miR-195-5p, miR-203a-3p	Breast cancer	Induces CSC phenotype; upregulates stemness genes	Targets ONECUT2	[96]
	miR-328-3p	Ovarian CSC	Maintains stemness	Suppresses DNA damage-binding protein 2; low ROS enhances ERK signaling	[97]
	miR-30b-3p	Glioma CSC (hypoxia)	Anti-apoptosis; chemoresistance	Targets RHOB; reduces apoptosis	[98]
	miR-146a-5p	Bladder CSC	Maintains stemness; enhances chemoresistance	Regulates cell cycle and apoptosis pathways	[99]
	miR-210	Pancreatic CSC (gemcitabine-resistant)	Confers drug resistance	Upregulates MDR1, YB-1, BCRP; activates mTOR	[100]
	miR-221	Colorectal CSC	Promotes stemness and chemoresistance	Targets QKI mRNA 3′UTR	[101]
	miR-155	Breast CSC	Induces EMT and chemoresistance in sensitive cells	Downregulates C/EBP-β; inhibits TGF-β, C/EBP-β, FOXO3a	[102]
	miR-485-5p	Oral CSC	Maintains stemness and chemoresistance	Suppresses KRT17; regulates integrin-mediated FAK/Src/ERK signaling and β-catenin nuclear translocation	[103]

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
