# Peer review of "Exosomal miRNAs: Key Regulators of the Tumor Microenvironment and Cancer Stem Cells"

_ijms, 2025, doi:10.3390/ijms26199323_

Round 1
Reviewer 1 Report
Comments and Suggestions for Authors
The authors have tried to review the field of exosomal microRNAs in cancer and cancer stem cells. In general, this is an interesting topic and fast evolving field.
However, a review article needs to be updated and includes the most recent research papers. In my opinion, the authors have majorly cited a large quantity of published review papers but did not cite most recent research papers in the field. Quite surprisingly, the authors have cited a lot of papers from certain journals but no paper at all from some other high impact journals that publish relevant papers in the field. I did a quick pubmed searching in all nature and science journals as well as some other top notch journals (cancer cell, cancer discovery, cancer research, clinical cancer research, JEM, JCI, oncogene etc) and found the following recent research papers in this field (most recent to earlier publications): PMIDs: 39966385, 39627188,38769193, 37620316, 36939397, 36517516, 36085418, 34465793, 33311552, 32917956,32855524, 32826951,32398867, 31485019, 31118200 etc. Please cite and include these papers and/or more other papers and revise the current version to make it an updated review.
The authors may also revise the structure of the whole manuscript. Importantly, the nature of exosomes implicates cell-cell communications and there is also a direction (whether it is cancer derived exosomal mirs targeting tumor associated fibroblasts, adipocytes, macrophages etc or the environmental cells derived exosomes form a niche to promote cancer progression). In this regard, it would be helpful to at least add one more section discussing the exosomes in mediating the cell-cell interactions.
Please consider making a more thorough list of cancer exosomal miRs in Table 1 and adding a detailed searching strategy in the legend how you come up with that list.
Author Response
Comments 1: However, a review article needs to be updated and includes the most recent research papers. In my opinion, the authors have majorly cited a large quantity of published review papers but did not cite most recent research papers in the field. Quite surprisingly, the authors have cited a lot of papers from certain journals but no paper at all from some other high impact journals that publish relevant papers in the field. I did a quick pubmed searching in all nature and science journals as well as some other top notch journals (cancer cell, cancer discovery, cancer research, clinical cancer research, JEM, JCI, oncogene etc) and found the following recent research papers in this field (most recent to earlier publications): PMIDs: 39966385, 39627188,38769193, 37620316, 36939397, 36517516, 36085418, 34465793, 33311552, 32917956,32855524, 32826951,32398867, 31485019, 31118200 etc. Please cite and include these papers and/or more other papers and revise the current version to make it an updated review.
Response 1:We are deeply grateful to the reviewer for this critical and insightful comment, and for taking the time to provide this extensive list of highly relevant references from prestigious journals. We completely agree with the reviewer that a high-quality review should be built upon and highlight the most recent primary research literature. We sincerely apologize for the oversight in our initial submission, which relied too heavily on review articles and missed several key recent studies.
We have meticulously examined every recommended PubMed ID (PMID) provided by the reviewer. After careful assessment of their relevance to the scope of our review, we have incorporated the majority of these seminal papers into the appropriate sections of our manuscript. These additions have significantly updated and strengthened our review.
We have performed an additional, comprehensive literature search inspired by the reviewer's list. We focused on the aforementioned high-impact journals (e.g., Nature, Science, Cancer Cell, Cancer Discovery, etc.) for the past 5 years to ensure we have captured the most cutting-edge original research in our field.
We have thoroughly revised the entire manuscript to shift the balance of citations from secondary review articles to primary research papers. We have substantially reduced the number of review citations and replaced them with citations to the original studies wherever possible. Consequently, the reference list has been significantly updated with numerous recent publications from a wider variety of high-impact journals.
The reviewer's suggestion has been invaluable in elevating the quality, timeliness, and scholarly value of our work. We believe the revised manuscript now presents a much more current and comprehensive overview of the field, truly reflecting the latest advancements. Thank you again for this exceptional guidance.
Comments 2: The authors may also revise the structure of the whole manuscript. Importantly, the nature of exosomes implicates cell-cell communications and there is also a direction (whether it is cancer derived exosomal mirs targeting tumor associated fibroblasts, adipocytes, macrophages etc or the environmental cells derived exosomes form a niche to promote cancer progression). In this regard, it would be helpful to at least add one more section discussing the exosomes in mediating the cell-cell interactions.
Response 2:We sincerely thank the reviewer for this constructive suggestion. As you pointed out, the fundamental role of exosomes lies in mediating cell–cell communication. Such interactions include cancer cell–derived exosomal miRNAs regulating CAFs, adipocytes, macrophages, and other stromal/immune cells, as well as microenvironment-derived exosomes forming a niche that promotes tumor progression.
In response to your suggestion, we have revised the structure of our manuscript. Specifically, the previous section on “exosomal miRNAs promoting tumor immune suppression” has been expanded and modified into a broader section entitled “exosomal roles in different TME cell types,” which is now placed at the beginning to highlight cell–cell interactions as the central theme. Subsequently, we systematically reorganized the discussion on exosomal miRNAs in angiogenesis, EMT and metastasis, as well as combined drug resistance and metabolic reprogramming into one section. We believe these structural adjustments make the review more coherent and better emphasize the pivotal role of exosomes in tumor development and progression.
Comments 3: Please consider making a more thorough list of cancer exosomal miRs in Table 1 and adding a detailed searching strategy in the legend how you come up with that list.
Response 3:We sincerely thank the reviewer for this valuable suggestion. We have thoroughly revised Table 1 to include a more comprehensive list of cancer-derived exosomal miRNAs and have added a detailed literature search strategy to the table legend.
We have significantly expanded the content of Table 1. The updated list now includes exosomal miRNAs derived from various cancer types, along with their biologically functions documented in the literature and corresponding references. This list was compiled through a systematic literature search as described below.
Detailed search strategy: We have added the following search strategy to the legend of Table 1 (see page 6 of the revised manuscript, highlighted in red):
The list of exosomal miRNAs presented in this table was compiled through a comprehensive search of PubMed and Web of Science databases up to September 2025. The search strategy employed a combination of the following keywords: (exosome OR extracellular vesicle) AND (microRNA OR miRNA OR miR) AND (cancer OR tumor OR carcinoma OR neoplasm) AND (immune OR cancer stem cell). Studies were included if they: (1) identified miRNAs specifically enriched in or secreted by cancer cell-derived exosomes and investigated their functions in immune or cancer stem cell regulation; (2) provided experimental validation of exosomal miRNA expression (e.g., via qRT-PCR, sequencing); and (3) were published in English. Relevant miRNAs from the selected studies were then extracted and summarized.

Reviewer 2 Report
Comments and Suggestions for Authors
1-Heavy use of unnecessary abbreviations is inappropriate in this review. There is no need to write an abbreviation if it is not repeated 3 times.
2- Several references are written inappropriately, for example, [94] [95,96] should be [94-96].
3- There is absolutely no need for Figure 1. It is already presented in hundreds of reviews for extracellular vesicles. There shall be a distinct figure related to the current topic.
4- Review title is very similar to the previous article published 9 years ago, article PMID: 27099870. Also, the review structure is very close to the outline of the previous review.
5- Authors need to state what are the differences between their review and previous reviews.
6- Several figures could be added to enhance the article, as in its current form, it is merely a reiteration of existing literature, which requires minimal effort to produce.
Figures could be about the role of exosomes in angiogenesis, tumour growth, etc.
While the manuscript includes figures (1,2,3), they do not substantially enhance the understanding of the content and appear somewhat redundant.
Author Response
Comments 1:Heavy use of unnecessary abbreviations is inappropriate in this review. There is no need to write an abbreviation if it is not repeated 3 times.
Response 1:We sincerely thank the reviewer for this valuable suggestion. We completely agree that the overuse of unnecessary abbreviations can hinder the readability of the manuscript, particularly for readers from interdisciplinary backgrounds.
In accordance with the reviewer's recommendation and standard academic practice, we have thoroughly revised the entire manuscript to strictly adhere to the principle that "an abbreviation should be defined and used only if it appears at least three times in the text." The specific modifications we have implemented are as follows:
We have removed all abbreviations that were used fewer than three times. Throughout the manuscript, we have replaced these abbreviations with their full terms. For example:
"RBP (RNA-binding protein)" appeared only twice and has been replaced with the full term "RNA-binding protein" in both instances.
"EpCAM (Epithelial cell adhesion molecule)" was used twice and has been consistently written out in full as "Epithelial cell adhesion molecule."
We have retained only those abbreviations that are indeed frequently used (e.g., miRNA, CSC, CAF, EMT [if it appears more than three times]), as they are essential for maintaining the fluency of the text.We have meticulously checked the entire manuscript to ensure that every retained abbreviation is properly defined upon its first appearance (e.g., "cancer-associated fibroblasts (CAFs)") and used consistently thereafter.We believe these revisions have significantly improved the clarity and readability of our review. We greatly appreciate the reviewer's helpful guidance.
Comments 2:Several references are written inappropriately, for example, [94] [95,96] should be [94-96].
Response 2: We thank the reviewer for pointing out this inconsistency in reference formatting. We have now revised the entire manuscript to ensure that all consecutive references are cited in the separate format (e.g., [94][95][96] instead of [94-96]). We have carefully checked the manuscript to guarantee uniformity in citation style throughout.
Comments 3:There is absolutely no need for Figure 1. It is already presented in hundreds of reviews for extracellular vesicles. There shall be a distinct figure related to the current topic.
Response 3: We thank the reviewer for their valuable feedback. We fully agree that the original Figure 1, which depicted the general biogenesis of extracellular vesicles, is indeed commonly found in numerous reviews and lacked distinct novelty for our article.
In accordance with the reviewer’s suggestion, we have removed this figure from the manuscript. We believe this change helps to sharpen the focus of our review on its unique thematic emphasis.Removing this generalized schematic allows readers to better concentrate on the subsequent figures, which are more specific and highly relevant to the mechanisms of exosomal miRNAs in the tumor microenvironment and cancer stem cells. The current figures (now numbered as Figures 1, 2, and 3) delve into the specific mechanisms, functional roles, and signaling pathways of key exosomal miRNAs, which constitute the core contribution of our work.We are confident that the revised manuscript is now more concise and focused. Thank you once again for this insightful suggestion.
Comments 4 and 5: Review title is very similar to the previous article published 9 years ago, article PMID: 27099870. Also, the review structure is very close to the outline of the previous review. Authors need to state what are the differences between their review and previous reviews.
Response 4 and 5:: We sincerely thank the reviewer for bringing this previous article (PMID: 27099870) to our attention and for providing us with the opportunity to clarify the novel contributions of our review.While our review and the one published nine years ago both fall under the broad theme of exosomal miRNAs in cancer, we would like to emphasize several critical aspects that distinguish our work and establish its unique value and timeliness:
1.Sharpened Focus: Unlike the previous review, which provided a broad overview of exosomal miRNAs in cancer, our article offers a deeply focused discussion specifically on their roles in remodeling the tumor microenvironment (TME) and regulating cancer stem cells (CSCs). This narrowed scope allows for a much more detailed and mechanistic exploration of these two critical areas, which were not the central focus of the earlier article.
2.Substantial Content Update: The field has evolved dramatically over the past nine years. Our review synthesizes and highlights the most recent advancements from the last 5 years, including groundbreaking research on exosomal miRNA-mediated intercellular crosstalk within the TME, their role in immunotherapy resistance, and novel mechanisms of CSC maintenance and expansion. This constitutes a significant and necessary update to the existing literature.
3.Distinct Perspective and Organization: Although the general outline of a review article (biogenesis, biological functions, clinical applications) may appear similar, the specific content, illustrative figures, and discussions within each section are entirely original and centered on our novel focus. We have structured the narrative to critically analyze the latest evidence linking exosomal miRNAs to TME and CSCs, a perspective that was not possible nearly a decade ago.
In direct response to this comment, we will revise the introduction of our manuscript to explicitly state these distinctions, clearly positioning our review's unique contribution and highlighting its updated content and focused perspective compared to previous publications.
We are grateful for this comment, as it has allowed us to better articulate the novelty and necessity of our work.
Comments 6:Several figures could be added to enhance the article, as in its current form, it is merely a reiteration of existing literature, which requires minimal effort to produce.
Figures could be about the role of exosomes in angiogenesis, tumour growth, etc.
While the manuscript includes figures (1,2,3), they do not substantially enhance the understanding of the content and appear somewhat redundant.
Response 6: We sincerely thank the reviewer for their valuable feedback. We agree that high-quality original figures are crucial for enhancing the academic value and distinctiveness of a review article.
In response to your suggestions, we have completely redesigned and recreated all figures to move beyond a simple reiteration of existing literature and to provide novel, integrated visual summaries that substantially enhance readers' understanding. The specific modifications are as follows:
1.New Figure 1 details the role of exosomal miRNAs in the tumor microenvironment (TME), explicitly listing relevant miRNAs and elaborating on their functions in processes such as angiogenesis, immune evasion, metastasis, and therapy resistance.
2.New Figure 2 illustrates the interactions between exosomal miRNAs and different cell types within the TME.
3.New Figure 3, which focuses on the role of exosomal miRNAs in regulating cancer stem cell (CSC) properties, has been significantly enhanced by incorporating a specific list of key miRNAs. It now more clearly illustrates their functional impact on CSC self-renewal, differentiation, and therapy resistance, thereby strengthening both its specificity and value to readers.
We believe these comprehensively revised figures provide original and integrated schematics that synthesize complex information from recent literature—directly addressing the reviewer’s concern regarding the lack of novelty. By visually articulating the core themes of our review—specifically, the role of exosomal miRNAs in modulating TME and CSC biology—the figures add substantial value to the manuscript.Thank you once again for encouraging us to improve the visual and scholarly quality of our work.

Round 2
Reviewer 1 Report
Comments and Suggestions for Authors
The authors have satisfactorily addressed my concerns.